# Exploring U.S. Food System Workers’ Intentions to Work While Ill during the Early COVID-19 Pandemic: A National Survey

**DOI:** 10.3390/ijerph20021638

**Published:** 2023-01-16

**Authors:** Caitlin A. Ceryes, Jacqueline Agnew, Andrea L. Wirtz, Daniel J. Barnett, Roni A. Neff

**Affiliations:** 1Department of Health Sciences, Towson University, Towson, MD 21252, USA; 2Department of Environmental Health & Engineering, Johns Hopkins Bloomberg School of Public Health, 615 N. Wolfe Street, Baltimore, MD 21205, USA; 3Department of Epidemiology, Johns Hopkins Bloomberg School of Public Health, 615 N. Wolfe Street, Baltimore, MD 21205, USA; 4Johns Hopkins Center for a Livable Future, 111 Market St., Ste. 840, Baltimore, MD 21202, USA

**Keywords:** safety climate, disaster preparedness, presenteeism, food system, worker, food insecurity, COVID-19

## Abstract

With “stay at home” orders in effect during early COVID-19, many United States (U.S.) food system workers attended in-person work to maintain national food supply chain operations. Anecdotally, many encountered barriers to staying home despite symptomatic COVID-19 illness. We conducted a national, cross-sectional, online survey between 31 July and 2 October 2020 among 2535 respondents. Using multivariable regression and free-text analyses, we investigated factors associated with workers’ intentions to attend work while ill (i.e., presenteeism intentions) during the early COVID-19 pandemic. Overall, 8.8% of respondents intended to attend work with COVID-19 disease symptoms. Almost half (41.1%) reported low or very low household food security. Workers reporting a higher workplace safety climate score were half as likely to report presenteeism intentions (adjusted odds ratio [aOR] 0.52, 95% confidence interval (CI) 0.37, 0.75) relative to those reporting lower scores. Workers reporting low (aOR 2.06, 95% CI 1.35, 3.13) or very low (aOR 2.31, 95% CI 1.50, 3.13) household food security levels had twice the odds of reporting presenteeism intentions relative to those reporting high/marginal food security. Workplace culture and safety climate could enable employees to feel like they can take leave when sick during a pandemic, which is critical to maintaining individual and workplace health. We stress the need for strategies which address vulnerabilities and empower food workers to make health-protective decisions.

## 1. Introduction

After the 11 March 2020 World Health Organization COVID-19 pandemic declaration [1], the United States (U.S.) government deemed food system workers, i.e., those responsible for producing, processing, distributing, selling, and serving food, “essential” [2]. To maintain operation of the national food supply chain, many U.S. food workers attended work in person while “stay at home orders” were in effect during the early COVID-19 pandemic. Consequently, essential food workers experienced high levels of COVID-19 exposure and illness risks [3], largely due to inability to socially distance while working [4,5,6]. Presenteeism, a phenomenon wherein employees attend work despite symptomatic illness [7], is an important risk factor for workplace and community COVID-19 spread [8,9], especially in workplaces with limited social distancing. Previous research has found that the intent to work while ill (here termed “presenteeism intentions”) is associated with actual presenteeism behaviors [10]. Anecdotal evidence suggests that many food system workers encountered barriers to staying home if ill [11]. This study explores factors associated with workers’ presenteeism intentions to identify opportunities for preventing presenteeism, and therefore reducing workplace spread of COVID-19 and other infectious illnesses.

U.S. Food System Workforce: In the U.S. food system, approximately 21.5 million workers produce, process, distribute, sell, and serve food in mostly “non-relocatable” jobs [12,13]. Appendix A provides food sector and subsector characteristics. Despite doing diverse tasks across sectors and jobs, many food workers share demographic and occupational similarities. Additionally, all of these workers jointly contribute to maintaining the food supply chain. Studying these workers as a group rather than in occupational silos provides insights relevant to this large cohort, their individual sectors, and food system functionality and resiliency.

Even before the COVID-19 pandemic, many food system workers experienced challenges associated with negative outcomes [14,15,16], including presenteeism [17]. Food system jobs are often characterized by: full-time wages at or below the poverty line ([18,19,20], Appendix A); low unionization rates, job insecurity, and at-will employment [21]; precarious tipped work [13] or piece work [22]; and lacking sick pay and health insurance [23]. These jobs exhibit high injury and illness rates relative to national averages, despite suspected widespread reporting suppression [24,25] and surveillance exemptions [26]. Many food jobs exist in the “gig economy”, meaning they are commonly exempted from many labor protections [27] and occupational health surveillance [28].

*Presenteeism*: Early presenteeism research examined economic and productivity losses resulting from employees working while sick or injured [29]. More recently, studies have investigated the implications of presenteeism for food safety [30] and for worker and community health [7,31]. Pre-pandemic studies found that organizational factors (e.g., work policies or cultures), job characteristics (e.g., shift design, job demands), and personal characteristics (e.g., financial stability concerns, personal sense of duty, and perceived co-worker expectations) [17] can potentiate presenteeism.

A limited literature explores presenteeism among food system workers, identifying associations between presenteeism and high work demands; poor employer-employee communication; poor staffing; inadequate workplace policies (e.g., lacking paid sick leave or requiring doctors’ notes) [30,32,33]; poor workplace safety climate [34]; job insecurity, job dissatisfaction, and hazardous working conditions [35]. During the COVID-19 pandemic, one study of restaurant workers has found that expanding paid sick leave at a large restaurant chain reduced presenteeism when compared to similar chains [36]. Other studies among the non-healthcare worker cohorts suggest that COVID-19 presenteeism is associated with household income, food security, and age [37], poor access to health benefits [37], and poor workplace safety climate [38]. Despite their importance for maintaining national food security, their high occupational vulnerability, and concerns about COVID-19 spread, little is known about how food system workers navigated decisions to attend work if ill during the early COVID-19 pandemic.

COVID-19 Presenteeism-Related Policies: At the time of survey, COVID-19 case rates and deaths were rising [39] and the Centers for Disease Control and Prevention (CDC) had issued guidance for sick workers to stay home or isolate [40]. However, concerns remained that exacerbated financial pressures and other factors could incentivize presenteeism [41,42]. In April 2020, the federal government implemented the first national sick leave policy [43] and augmented unemployment insurance [44]. The former provided paid sick leave for many food chain workers who had previously lacked this benefit, including part-time workers [43,45]. However, firms employing fewer than 50 or more than 500 people were excluded from this policy, and voluntary implementation was inconsistent [36]. Additionally, many processing workers were encouraged or required to work with COVID-19 symptoms [6,46] following a presidential executive order preventing closures of meat and poultry processing plants [47].

To our knowledge, no study has examined presenteeism intentions in a large, nation-wide, food system worker sample. Here we explore workplace and non-workplace factors associated with food system worker COVID-19 presenteeism intentions during the early COVID-19 pandemic to identify opportunities to support food workers to remain home if ill or at risk of infecting others.

## 2. Materials and Methods

We drew data from the Johns Hopkins COVID-19 Food Worker Survey, developed during the early COVID-19 pandemic and deployed from 31 July 2020 to 2 October 2020. This cross-sectional, national, online survey of 3399 food system workers documented COVID-19 pandemic-related workplace experiences and conditions. Recruitment and survey design have been reported in depth elsewhere [3].

Study population: The survey included individuals who worked in any of six targeted food system sectors (production; processing; distribution; retail; service; assistance), who were literate in English or Spanish, who lived in the U.S., who were 18 years old or older, and who had attended a food system job in-person since 11 March 2020.

Of the 3831 who initiated the survey, 25.4% of the respondents did not answer the outcome question corresponding to presenteeism intentions; thus, their data were excluded from analyses. We also excluded participants who had previously contracted COVID-19, and/or did not receive a paycheck, producing an analytic sample of 2535 participants. Participants missing outcome data were more likely to identify as Hispanic/Latinx and/or work at organizations with fewer than 10 employees than those with outcome data. Missing outcome data was not associated with age, race, gender, U.S. census region, having worked in the past month, or degree of customer interaction. We discuss missing data patterns for independent variables and implications for interpretations in the discussion.

Sample size calculations determined that a sample of at least 1000 respondents would provide enough power to detect group differences using a 3% margin of error and 95% confidence for the outcome. The median survey completion time was 19.5 min.

Instrument: In brief, the 114-item instrument was created with input from workers and worker representatives and experts in survey design, disaster preparedness, and occupational health. We used validated scales where possible and included novel items to capture COVID-19-related perspectives about working conditions. Measures are summarized below.

Measures: Demographics included age, gender identity, race, ethnicity, highest educational attainment, household income, and geographic location. All questions included “don’t know” or “not applicable” options and participants could skip any item beyond demographics. The survey was terminated if demographic responses did not satisfy inclusion criteria. Appendix A presents survey items and coding.

Presenteeism Intentions: We derived our main outcome from the level of agreement with the statement: “If I was sick with COVID-19, but I was still able to work, I would go to work”. The 5-point Likert scale was dichotomized to: workers who strongly agreed or agreed with the statement versus all others. As few COVID-19-specific survey items existed early in the pandemic, we crafted this item based on questions from existing disaster preparedness literature assessing hospital workers’ willingness to work during disaster scenarios, including pandemic influenza [48].

Occupational Measures: Workers indicated their food system sector and subsector from an edited Food Chain Workers Alliance list (FCWA; a coalition of food worker-based organizations; [13]). Workers employed in more than one sector were asked to indicate the job in which they worked the most hours. Occupational characteristics included job tenure, full/part-time status, organization size, customer contact, work transportation, whether workers were told they were “required” to work by their employers, and union membership. Respondents specified all workplace benefits provided by their employers since the pandemic declaration from a select-all-that-apply list [49]. These were aggregated as frequencies and analyzed individually. We assessed quantitative work demands and workplace social support using medium-length scales from the Copenhagen Psychosocial Questionnaire III (COPSOQ III) [50], following published scoring procedures and then dichotomizing scores at the median into “high” and “low” categories. Higher work demands scores indicated more challenging levels of work demands (e.g., time pressure or many overlapping tasks). We assessed organizational safety climate using a 6-item scale [51] where high scores indicated that workers perceived their organization had a high commitment to safety. We created a composite organizational safety climate variable by summing scale responses and dichotomizing at the median, including responses for all participants who had answered 5 or more (of 6) items.

Non-Occupational Measures: We measured food security since 11 March 2020, using a United States Department of Agriculture (USDA) Six-item Short Form Household Food Security Survey Module [52]. The composite categorical variable included responses of participants with 2 or more items (of 6) and was scored according to USDA classifications: high/marginal food security (raw score 0–1); low food security (2–4); and very low food security (5–6). Cronbach’s alpha was >0.7 for all scales except quantitative work demands, which was 0.67 [53].

We measured attitudes regarding reopening the economy based on agreement with the statement, “It is worth the health risk to reopen the economy as soon as possible”. The 6-point Likert scale was collapsed to 3 points: agreement; neither agreement nor disagreement; and disagreement.

Theoretical Approach: The Job Demands-Resources (JD-R) [54] and Total Worker Health (TWH) models [55,56] guided analyses. The JD-R model suggests that job resources can mitigate the negative health impacts of workplace demands [54]. We therefore hypothesized that resources such as organizational safety climate (defined as employees’ shared perceptions of their organization’s prioritization of worker safety [51,57]) and paid sick leave would reduce the likelihood of workers reporting presenteeism intentions. The Total Worker Health approach [56] considers external (i.e., non-workplace) factors that impact worker well-being. Our conceptual model (presented in Ceryes et al., 2021 [3]) includes workplace and non-workplace factors associated with food worker outcomes, including presenteeism, during the COVID-19 pandemic.

Statistical analyses: We used STATA 14 I/C (College Station, TX, USA) for quantitative analyses. Statistics included Chi^2^ or Rank Sum tests (significance value *p* < 0.05) as well as Spearman’s rank and Pearson’s correlation coefficients to identify collinearity. We used bivariate logistic regression to assess correlations according to presenteeism intention status. Adjusted logistic regression models were used to examine associations with workplace characteristics. Variables associated with the outcome, presenteeism intentions, at the level of *p* < 0.05 were retained in the multivariable model. These were age, gender, food system sector, organization size, hourly status.

Additional covariate inclusion was informed by *a priori* conceptual associations (race, ethnicity, geographic location). We included food security status and perspectives on reopening the economy based on free-text data (described below) and bivariable associations (*p* < 0.05). The final model estimated associations between presenteeism intentions, workplace, and non-workplace characteristics while controlling for age, race, ethnicity, gender, food system sector, organization size, and hourly status. Akaike’s Information Criteria (AIC) values were used to assess model fit, and variance inflation factors assessing multicollinearity were all less than four (mean = 1.43) [58].

Sensitivity analyses were conducted by stratifying on degree of customer interaction and whether workers were told they were “required” to work. We also controlled for clustering at the state level. Estimates did not meaningfully differ from our primary results (Appendix A).

Free-text Analyses: Many survey participants provided detailed responses to the open-ended question: “Do you have any other comments about the level of risk from COVID-19, or decisions about whether to go to work?”. These comments often included discussion of presenteeism intentions; thus, we analyzed responses to elaborate on our quantitative findings [59]. This approach has been used previously in survey-based presenteeism studies [60]. Comments informed covariate selection by narrowing variables considered for analyses. For example, responses frequently mentioned food insecurity and perspectives on opening the economy; therefore, we retained those variables. We also used comments to choose between highly correlated variables (e.g., food security status over annual household income). Finally, the free text results informed interpretation and discussion of quantitative results.

The lead investigator (CAC) conducted two close reviews of free-text data, taking notes before coding responses and organizing them into themes [61], and excluding non-substantive comments (e.g., “N/A” or “No”). Atlas.ti (Version 8.0, Berlin, Germany) and Microsoft Excel (Washington, DC, USA) were used to sort, organize, and manage free-text data. Respondents offering comments were compared to those who did not and to the full sample to identify potential biases. We analyzed presenteeism-related text responses overall and by sector, by subgroups according to reports of presenteeism intentions or behaviors, and by benefits and working conditions. Qualitative memos tracked CAC’s reactions to comments [62].

The Johns Hopkins Bloomberg School of Public Health Institutional Review Board considered this study exempt (category 2) (IRB No. 12549).

## 3. Results

### 3.1. Quantitative Results

Table 1 presents analytic sample demographics. Respondents were primarily female (64.8%), not Hispanic/Latinx (90.0%), white (86.0%), non-union (79.6%), working full-time (64.8%) and of average age 45.9 years (SD 11.2). Most worked in restaurant/service (43.3%) and retail (34.9%), with the fewest in distribution (2.4%). Almost all (95.9%) had worked in-person in the past month before taking the survey. Nearly a third (32.7%) were told they were “required” to work by their employers at some point between pandemic onset and the survey in August–September 2020. Almost half of respondents (41.1%) reported low or very low food security. Analytic sample demographics resembled those of the overall study population.

Presenteeism: Of 2535 respondents, 8.8% agreed that they would attend work if sick with COVID-19, but these differed greatly by sector. Table 2 provides an overview of outcome prevalence. Appendix A provides group comparisons between groups reporting presenteeism intentions versus not by variables of interest.

Benefits: Of 2527 respondents, 27.7% reported paid sick leave access, and 30.1% reported “easier” access to sick leave since 11 March 2020. Fourteen percent reported that they had received free workplace COVID-19 testing since the pandemic declaration.

Multivariable Model: Table 3 presents bivariate (Model 1) and multivariable logistic regression (Models 2 and 3) results for variables of interest (organizational safety climate; work demands; access to paid leave; food security; perspectives about reopening the economy) and presenteeism intentions. These were adjusted for age, gender, ethnicity, race, full/part-time status, food system sector, and organization size. See Appendix A for all models.

After adjustment, respondents reporting high levels of organizational safety climate were half as likely to report presenteeism intentions, compared to those reporting lower scores (adjusted odds ratio [aOR] 0.52, 95% CI 0.37, 0.75). Workers with high levels of work demands had 49% greater odds of reporting presenteeism intentions relative to those reporting lower levels (aOR 1.49, 95% 1.03, 2.16). Food production workers had higher odds of reporting presenteeism intentions relative to retail workers (aOR 3.96; 95% CI 1.98, 7.92). Paid sick leave was not associated with presenteeism intentions.

Respondents reporting low or very low food security were more than twice as likely to report presenteeism intentions relative to those reporting marginal/high food security (aORs 2.06, 95% CI 1.35, 3.13 and 2.31, 95% CI 1.50, 3.13, respectively). Respondents who agreed or strongly agreed that it was “worth the health risk” to reopen the economy had higher odds of reporting presenteeism intentions relative to those who disagreed with this statement (aOR 2.43, 95% CI 1.58, 3.73).

### 3.2. Free Text Results

Overall, 13.5% of respondents answered the question, “Do you have any other comments about the level of risk from COVID-19, or decisions about whether to go to work?” and 460 comments were substantive. Responses ranged from 1 to 233 words, with 23-word median length. Production workers had the lowest median word count (13 words) and retail the highest (24 words). Workers who commented were less likely to work in food production or report annual household incomes below $15,000 or above $100,000. Workers who commented were more likely to work for tips and report very low food security status than those who did not (Appendix A). Table 4 provides illustrative quotations from free-text data, organized by themes and sub-themes.

#### 3.2.1. Workplace Factors

Policies: Many comments mentioned employers’ policies relating to presenteeism and workplace COVID-19 spread. While a few workers described adequate sick pay if symptomatic or COVID-19- positive, others described insufficient policies and benefits, including lacking paid sick leave. Respondents also described barriers to quarantine and testing, including financial disincentives for disclosing COVID-19 exposure, unpaid isolation periods, and high test costs. Others described policies providing only partial sick pay, or policies requiring employees to find shift coverage, use personal vacation time, obtain doctors’ notes, or abide penalty-driven attendance systems.

Culture: Even if employers had official policies supporting those who stayed home, employees described cultural factors which communicated an expectation to work even if symptomatic with COVID-19. Many workers expressed concerns about following anti-COVID-19 guidelines stemming from high levels of perceived job insecurity. For example, workers cited concerns about employer retaliation for using sick leave. Other comments described instances where policies meant to discourage COVID-19 presenteeism were unclear or not followed. Examples include instances of symptomatic co-workers continuing to work following symptom-checks, and managers ignoring COVID-19 symptoms rather than sending staff home.

#### 3.2.2. Non-Workplace Factors

Economic precarity: Aside from workplace conditions, workers cited economic instability, stemming from insufficient wages, as a driver for presenteeism. Many comments mentioned the need to make ends meet, working paycheck to paycheck, and working to buy food for workers’ families.

*Distrust of public health messaging*: Some respondents viewed COVID-19 disease risks as exaggerated or not a credible health threat and indicated this perspective would influence their decisions to attend work with COVID-19 symptoms.

## 4. Discussion

Our findings identify workplace and non-workplace conditions associated with food system workers’ intentions to work while ill and provide insights into this decision. While our results are specific to the COVID-19 pandemic context, we believe they have relevance for both infectious disease outbreak planning and mitigating the spread of more quotidian contagions.

Given rapid changes in infection rates, resources available for worker protection, and scientific knowledge about COVID-19 throughout 2020 and 2021, it is important to view these results in their temporal context. This study occurred during the first four to six months of the pandemic. At this time, vaccines were unavailable, federal paid sick leave policies had been enacted, and eviction moratoriums and unemployment insurance enhancements were in place [63]. Because of rapid U.S. case-rate increases and news coverage emphasizing disease severity during these months [64], respondents may have perceived COVID-19 as more severe than other illnesses and planned to remain home. As the pandemic continued, many states prioritized “reopening”. Essential and non-essential workers were encouraged to return to work, and supporting policies were relaxed or rescinded. Therefore, if repeated later in the pandemic, a similar study might show an even greater prevalence of presenteeism intentions among these workers.

### 4.1. Workplace Factors Associated with Presenteeism Intentions

*Organizational Safety Climate*: Workers who received a high safety climate scale score perceived that their employers valued and prioritized their safety at work. These workers were substantially less likely to report COVID-19 presenteeism intentions. This finding aligns with other pre- and mid-pandemic studies suggesting that safety climate influences workers’ presenteeism decisions [38,65,66,67]. It also builds on previously established connections between safety climate and COVID-19 safety perceptions [3].

Organizational safety climate constructs include employees’ shared perceptions of safety priorities, policies, and procedures; managerial commitment to safety; employee behavioral norms, and worker safety activity participation [51]. Free-text data elaborated on how these constructs could influence presenteeism intentions. For example, comments describing managers ignoring COVID-19 safety policies could indicate a lack of employee empowerment to participate in safety activities and policy enforcement. This lack of empowerment could possibly extend to employees feeling that they could not stay home if ill.

Organizational safety climate is often studied regarding its effects on injury prevention, but these findings suggest its underlying constructs could represent important intervention targets for reducing illness-related presenteeism. Improving safety climate could work synergistically with other workplace culture components known to be associated with reducing presenteeism, such as having strong workplace social communities, especially in circumstances of work–life imbalance [68].

Sick Leave: The lack of association between sick leave access and presenteeism intentions after adjustment was surprising. Workers’ comments describing cultural and organizational barriers to using sick leave, even if it was “officially” established, provide one interpretation of this finding. Descriptions of retaliation and penalties barring workers from accessing sick leave indicate that some employees were not empowered to use it. Such barriers have been documented among restaurant workers [45], and we expand these findings to include other food system workers. Our results diverge from those of Schneider and colleagues’ (2021), who found that increasing paid sick leave reduced COVID-19 presenteeism among restaurant workers at the Olive Garden fast-casual restaurant chain. We suggest the difference could again relate to empowerment. Because Olive Garden’s paid sick leave expansion occurred following “significant public scrutiny”. [36], their employees might have felt more able to access their newfound benefits than workers whose employers were not being scrutinized.

Work requirements: Notably, 32.7% of respondents reported being told they were “required” to work during the COVID-19 pandemic. Because these workers lacked a choice, this circumstance would not typically be considered presenteeism. Sensitivity analysis estimates of reported presenteeism intentions, stratified by requirement to work, did not meaningfully differ from our primary results. Research should assess the physical and mental health impacts of requirements to work during the COVID-19 pandemic.

Sector differences: After controlling for demographics and job characteristics, production workers were more likely to report presenteeism intentions relative to retail workers. This finding could relate to reduced risk perceptions due to these workers’ open-air working environments and not typically interacting with customers. Alternatively, H-2A visa holders (meaning those in the United States on temporary agricultural work visas) might feel obliged to attend work while ill in order to remain in the country [69]. Research is needed to explore this association further. We did not identify other sector-specific differences or note differential comment content by sector, though production workers were less likely to provide comments than workers in other sectors.

### 4.2. External Factors Associated with Presenteeism Intentions

Food Security: Over 40% of respondents reported experiencing low or very low food security, despite working at in-person food jobs during the COVID-19 pandemic. After controlling for covariates, these workers were more than twice as likely to report presenteeism intentions than those with marginal or high food security. This finding, combined with many free-text comments that mentioned the need to work to buy food, suggests food insecurity was a major driver of presenteeism intentions in this population. Our findings align with Tilchin and colleagues’ (2021) findings that perceived food insecurity was associated with a three-fold increase in intention to work sick among U.S. employees. They also align with other studies which highlight connections between presenteeism and financial instability during COVID-19 [70]. The paradox of food workers experiencing food insecurity while feeding the nation has been previously acknowledged in literature on farmworkers [71], and we re-emphasize its inherent inequity here. We also note that these findings could help explain broader disparities in COVID-19 morbidity and mortality [72] during early pandemic waves.

Risk Perceptions: Workers who felt it was “worth the health risk” to reopen the economy were twice as likely to report presenteeism intentions. Comments suggested some respondents did not trust public health messaging about COVID-19’s severity, and/or felt the benefits of working, including financial stability, outweighed COVID-19 exposure risks. This finding highlights the importance of effective and consistent public health messaging for reducing infectious disease spread.

### 4.3. Future Research and Recommendations

This study provides evidence about self-reported presenteeism intentions, and future studies are needed to measure actual presenteeism behaviors related to both physical and mental illnesses in this population. Longitudinal studies should further examine the potential association between workplace culture and presenteeism, especially whether shifts in workplace safety climate can decrease the spread of workplace and community infectious disease. Research is also needed to explore ways to empower employees to fully participate in developing and enacting policies, such as paid sick leave and symptom checks, especially in the context of top-down federal or state policy mandates and prolonged emergencies or pandemics.

This study suggests that worker food insecurity represents a major driver of COVID-19 presenteeism intentions. We therefore endorse instituting and evaluating policies that improve workers’ overall financial stability to prevent presenteeism and accompanying disease transmission. These policies include raising food workers’ compensation to a living wage, limiting “just in time” shifts, standardizing work schedules so that workers can plan for childcare and other needs, and providing reliable, full-time, benefitted work to those who want it [73]. Such actions would not only contribute to public health and food system stability but could also reduce food businesses’ presenteeism-related economic losses, which are estimated to be substantial [29]. Finally, we advocate for heightened external accountability around workplace safety protocols and practices, including proactive worksite inspections and statutory worker protections, especially for “essential” workplaces. It would be informative to track presenteeism and its associated influences and outcomes in a longitudinal manner should a similar national disaster occur in the future.

### 4.4. Limitations

While this large national survey addresses the experience of a unique worker population that is critical to our food supply, there are some expected limitations. As with many other Internet-based surveys, our sample overrepresented white, female, and high-income individuals [74,75]. Despite efforts to minimize missing data, thus increasing sample size and diversity, few participants identified as African American and Hispanic/Latinx or other Black/Indigenous/People of Color (BIPOC) individuals. These groups are of great interest because they are believed to be more subject to the negative impacts of COVID-19 [76]. This study may have underestimated levels of risk factors or the existence of presenteeism intentions, especially among these populations. Future studies must focus on including these groups.

Use of free text data always presents the challenge of interpretation, especially when a single coder reviews the responses. However, our text analyses related directly to our validated scales and served the purpose of expanding, clarifying, and prioritizing those results.

This cross-sectional study was conducted during the early stages of the pandemic, when COVID-19 knowledge and risk perception were evolving and anxiety was high. Although the design does not allow for causal inferences, results during this critical period indicate participants’ perceptions of causal relationships between several risk factors and presenteeism decisions. Social desirability bias could have reduced respondents’ willingness to report presenteeism intentions, though data collection using an anonymous, Internet-based survey has been shown to reduce this bias [77].

## 5. Conclusions

The COVID-19 pandemic has highlighted U.S. society’s reliance on food system workers to maintain national food security. Despite their heightened risks for COVID-19 morbidity and mortality, many food system workers indicated they would attend work while ill during the early COVID-19 pandemic. Often, they felt that they had no choice. This research suggests that interventions targeting workplace safety climate and food insecurity among food system workers could reduce presenteeism, therefore protecting the national food supply and the public’s health during the COVID-19 pandemic and in other disaster or infectious illness scenarios. Addressing barriers to staying home when ill, such as improving safety climate and mitigating or eliminating vulnerabilities such as food insecurity, could enable food system workers to make decisions that protect both themselves and their workplaces. Reducing presenteeism is critical for creating optimal worker health outcomes, public health outcomes, and maintaining a functioning food system.

## Figures and Tables

**Table 1 ijerph-20-01638-t001:** Demographic and occupational characteristics for a national United States (U.S.) food system worker cohort during early COVID-19.

Demographic or Occupational Characteristic	*n* (%)
Age in Years	(*n* = 2535)
18–24	81 (3.2)
25–44	1054 (41.6)
45–65	1334 (52.6)
>65	66 (2.6)
Gender	(*n* = 2535)
Female	1641 (64.8)
Male	846 (33.4)
Other	48 (1.9)
Race	(*n* = 2527)
White	2196 (86.0)
African American	112 (4.4)
Other/Mixed race	242 (9.6)
Ethnicity	(*n* = 2440)
Not Hispanic/Latinx	2196 (90.0)
Hispanic/Latinx	244 (10.0)
Sector	(*n* = 2535)
Production	115 (4.5)
Processing	227 (9.0)
Distribution	60 (2.4)
Retail	884 (34.9)
Restaurant/Service	1097 (43.3)
Assistance	152 (6.0)
Household Income	(*n* = 2330)
<$25,000	642 (27.6)
$25,000–34,999	427 (18.3)
$35,000–49,999	427 (18.3)
$50,000–99,000	696 (30.0)
>$100,000	138 (5.9)
Food Security Status since pandemic declaration	(*n* = 2374)
High or marginal	1399 (58.9)
Low	505 (21.3)
Very low	470 (19.8)
Education	(*n* = 2353)
Up to/some high school	124 (5.3)
High school diploma/GED	789 (33.5)
Some college/associate degree	1104 (46.9)
Bachelor’s/ advanced degree	336 (14.3)
U.S. Census Region	(*n* = 2375)
Northeast	427 (18.0)
Midwest	654 (27.5)
South	857 (36.1)
West	437 (18.4)
Union Status	(*n* = 2471)
Non-Union Member	1965 (79.6)
Union Member	506 (20.5)
Employer Size	(*n* = 2454)
1–10	316 (12.9)
11–49	813 (33.1)
50–499	1120 (45.6)
>500	205 (8.4)
Hourly status	(*n* = 2332)
Full Time	1510 (64.8)
Part Time	651 (27.9)
Other	171 (7.3)
Worked in the last month	(*n* = 2535)
Yes	2430 (95.9)
No	105 (4.1)
Customer Contact	(*n* = 2523)
Yes	1918 (76.0)
No	605 (24.0)
Safety Climate Score	(*n* = 2375)
High	1069 (55.0)
Low	1069 (45.0)
Work Demands	(*n* = 2466)
High	1360 (55.2)
Low	1106 (44.9)
“Required” to work	(*n* = 2420)
Required to work during COVID-19	792 (32.7)
Asked to work but not required	623 (25.7)
Both required and asked at different times	324 (13.4)
Neither required nor asked	681 (28.1)

Percentages may not add to 100% due to rounding.

**Table 2 ijerph-20-01638-t002:** Prevalence of presenteeism intentions in a national sample of U.S. food system workers during early COVID-19, by food system sector (*N* = 2353).

Food System Sector	Workers Reporting Presenteeism Intentions *n* (%)
All sectors	222 (8.8)
Production	28 (24.4)
Processing	24 (10.6)
Distribution	8 (13.3)
Retail	66 (7.5)
Restaurant	91 (8.3)
Assistance	5 (3.3)
	*p* < 0.001

**Table 3 ijerph-20-01638-t003:** Workplace and non-workplace factors associated with reporting presenteeism intentions in a national food chain worker sample during early COVID-19.

	Model 1 ^+^	Model 2 ^++^	Model 3 ^+++^
	Odds Ratio	Odds Ratio	Odds Ratio
	95% CI	95% CI	95% CI
	*p* value	*p* value	*p* value
	*n*	*n*	*n* = 1793
Organizational Safety Climate Score			
Low	Ref	Ref	Ref
High	0.61	0.59	0.52
	0.46, 0.81	0.44, 0.79	0.37, 0.75
	0.001	<0.001	<0.001
	*N* = 2375	*N* = 2287	
Quantitative Work Demands			
Low	Ref	Ref	Ref
High	1.91	1.95	1.49
	1.42, 2.57	1.44, 2.65	1.03, 2.16
	<0.001	<0.001	0.03
	*N* = 2466	*N* = 2370	
Access to paid leave			
No	Ref	Ref	Ref
Yes	0.83	0.83	1
	0.60, 1.14	0.60, 1.14	0.67, 1.50
	0.25	0.25	0.99
	*N* = 2527	*N* = 2249	
Food Chain Sector			
Retail	Ref	Ref	Ref
Production	3.99	3.59	3.96
	2.43, 6.54	2.04, 6.34	1.98, 7.92
	<0.001	<0.001	<0.001
Processing	1.47	1.49	1.29
	0.90, 2.40	0.90, 2.46	0.67, 2.51
	0.13	0.12	0.45
Distribution	1.91	1.81	2.14
	0.87, 4.18	0.81, 4.05	0.88, 5.16
	0.11	0.15	0.09
Restaurant/Service	1.12	1.07	1.18
	0.81, 1.56	0.76, 1.51	0.72, 1.93
	0.5	0.7	0.51
Food Assistance	0.42	0.48	0.5
	0.17, 1.06	0.19, 1.23	0.14, 1.74
	0.07	0.13	0.28
	*N* = 2535	*N* = 2436	
USDA Food Security Category			
High	Ref	Ref	Ref
Low	2.33	2.31	2.06
	1.65, 3.29	1.61, 3.31	1.35, 3.13
	<0.001	<0.001	0.001
Very low	2.26	2.25	2.31
	1.59, 3.22	1.55, 3.24	1.50, 3.13
	<0.001	<0.001	<0.001
	*N* = 2374	*N* = 2282	
“It is worth the health risk to reopen the economy as soon as possible”			
Strongly/disagree	Ref	Ref	Ref
Neutral	1.29	1.28	1.44
	0.89, 1.87	0.87, 1.86	0.95, 2.16
	0.176	0.21	0.08
Strongly/Agree	2.27	2.23	2.43
	1.56, 3.30	1.51, 3.28	1.58, 3.73
	<0.001	<0.001	<0.001
	*N* = 2114	*N* = 2030	

^+^ Model 1: Unadjusted; ^++^ Model 2: Controlled for age, gender, race, ethnicity; ^+++^ Model 3: Controlled for age, gender, race, ethnicity, organization size, hourly status. Ref = Reference.

**Table 4 ijerph-20-01638-t004:** Illustrative quotations describing respondents’ perceptions of factors related to presenteeism intentions from a national sample of U.S. food system workers during early COVID-19.

Themes	Illustrative Quotations
**Workplace Factors**	
*Policies*	
Lacking sick pay for COVID-19 symptoms or exposures	“Obviously no one wants to go to work sick, but it is necessary since the pay is so low and I don’t get sick pay.” (Retail worker)
	“If I was to be exposed to someone with COVID I would not tell my [employers] about it because they will not pay me to be off work. I cannot afford to be off work”. (Retail worker)
	“… it is a 2 week or more wait for results. If you are tested you may not return to work until you get results. How many people with mild symptoms are going to be out of work for 2 weeks or more voluntarily?” (Restaurant worker)
Lacking financial support for testing	“The test cost as much as half of my weekly wage”. (Retail worker)
Punitive attendance policies	“If you were sick or had any of the symptoms of COVID-19, if you didn’t go to work they would “point” [penalize] you for that so if you have enough points you will eventually ‘point out’ [lose your job]”. (Processing worker)
*Culture*	
Fear of retaliation for using sick leave	“Calling in sick is frowned upon. People who call in sick frequently get less hours [meaning less pay] and the worse [less desirable] hours”. (Retail worker)
	“Even if you don’t get fired for calling out … they’ll find something else to fire you for…“. (Restaurant worker)
Employers discouraging use of anti-COVID-19 policies	“Boss told us not to get tested so we wouldn’t have to miss work”. (Retail worker)
**Non-workplace Factors**	
*Economic Precarity*	
Perceived food insecurity	“There is NO decision!… We have bills and children to feed…I cannot stay home!” (Processing worker)
	“What the **** am I gonna do, not feed my kids?… (pardon my profanity, it’s necessary for emphasis, I can’t really convey how strongly I feel about this)”. (Retail worker)
*Distrust of Public Health Messaging*	
Perceiving COVID-19 as a non-credible health threat	“I think it’s blown out of proportion and has very skewed and inaccurate testing. I don’t think I’m anymore at risk than the seasonal flu”. (Processing worker)

## Data Availability

Data from this study are available upon reasonable request.

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
