# Peer review of "Exploring U.S. Food System Workers’ Intentions to Work While Ill during the Early COVID-19 Pandemic: A National Survey"

_ijerph, 2023, doi:10.3390/ijerph20021638_

Round 1
Reviewer 1 Report
Very important topic to address in public health; there are multiple formating/grammar errors (e.g., inconsistent spacing, inappropriate use of hyphens, run-on sentences).
Author Response
Thank you for this helpful summary. We appreciate you noting the topic’s importance in public health. Indeed, there were errors in the manuscript, and we thank you for pointing them out. Please see the revised manuscript with these addressed.
Reviewer 2 Report
Comments for the authors:
This paper presents a theoretical model examining presenteeism in food system workers. This manuscript is providing valuable insight into this topic. In particular, it was very interesting to use free-text data for research. I suggest the following to make your research clearer and easier to understand.
Introduction
Because 'Presenteeism' is the result, and 'Presenteeism Intentions' is the situation before that, I think the two are different. Is there a special reason why you chose 'Presenteeism Intentions' as a variable instead of 'Presenteeism'? It is possible that presenteeism has already occurred in the COVID-19 situation. However, it is necessary to explain why the author wanted to see 'Presenteeism Intentions' rather than 'Presenteeism'.
Materials and Methods
I suggest that the Materials and Methods details are unified in format, such as Results and Discussion.
And it is necessary to present a reference to the 'Presenteeism Intentions' question. It appears to be a single item. Does this get at the issue of working while having mental health issues as well as physical health issues? And, how reliable is such self-report data? At the very least, more detail on reference use of it would help give the reader more confidence in its utility.
Results
Presenteeism is an important variable in this study in p.7. It will be easier to understand if the contents of Presenteeism are presented in a table.
I read free text results with great interest. However, there is a lot of information presented in text on p.9. Perhaps this would be easier to follow if presented in a figure or table.
Limitations
I suggest putting limitations in the separate section.
References
I suggest expanding the list. Maybe this you will find useful and interesting:
The Effects of Work Characteristics Related to Work–Life Imbalance on Presenteeism among Female Workers in the Health and Social Work Sectors: Mediation Analysis of Psychological and Physical Health Problems. International Journal of Environmental Research and Public Health, 18(12), 6218. https://doi.org/10.3390/ijerph18126218
Author Response
Thank you for your thoughtful and detailed review. We have attached point-by-point responses here.

Reviewer 3 Report
The paper presents a well-documented view of its central issue: presenteeism in the food industry. The authors argue the novelty of their survey based on a large work sample. They mentioned Appendices 1 and 2, but we don't have access. The text is clear, with a lot o pieces of evidence that are easy to interpret and do not require a piece of very specialized knowledge to reach conclusions. Attrition is treated correctly, with a sound explanation of the causes of the missing outcome data and its implications for the paper's conclusion.
I strongly suggest using the "Interactive clustering tree" approach to complement logistic analysis. The Interactive Clustering Tree (ICT) is an unsupervised clustering method (typology) that divides a population into clusters so that: • 2 cases from the same cluster resemble as much as possible, • 2 cases from distinct clusters differ as much as possible. The most notable feature of the ICT method is the ability to formalize the typology as a decision tree in one step. It makes it easier to analyze the results, improving discussion quality. But it is only a suggestion.
Author Response
Thank you for your time and help in reviewing our manuscript. Please see the attached comments.

Round 2
Reviewer 2 Report
Agreed. I recommend the publication in the present form.